# Oncologist phenotypes and associations with response to a machine learning-based intervention to increase advance care planning: Secondary analysis of a randomized clinical trial

**Eric Li**[1], **Christopher Manz**[2], **Manqing Liu**[1], **Jinbo Chen**[1], **Corey Chivers**[1], **Jennifer Braun**[1], **Lynn Mara Schuchter**[1], **Pallavi Kumar**[1], **Mitesh S. Patel**[1], **Lawrence N. Shulman**[1], **Ravi B. Parikh**[1] *

**1** University of Pennsylvania, Philadelphia, PA, United States of America, **2** Dana–Farber Cancer Institute/Massachusetts General Hospital, Jamaica Plain, MA, United States of America

* Ravi.Parikh@pennmedicine.upenn.edu

**Data Availability Statement:** All relevant data are within the paper and its Supporting Information files.

## Abstract

### Background

While health systems have implemented multifaceted interventions to improve physician and patient communication in serious illnesses such as cancer, clinicians vary in their response to these initiatives. In this secondary analysis of a randomized trial, we identified phenotypes of oncology clinicians based on practice pattern and demographic data, then evaluated associations between such phenotypes and response to a machine learning (ML)-based intervention to prompt earlier advance care planning (ACP) for patients with cancer.

### Methods and findings

Between June and November 2019, we conducted a pragmatic randomized controlled trial testing the impact of text message prompts to 78 oncology clinicians at 9 oncology practices to perform ACP conversations among patients with cancer at high risk of 180-day mortality, identified using a ML prognostic algorithm. All practices began in the pre-intervention group, which received weekly emails about ACP performance only; practices were sequentially randomized to receive the intervention at 4-week intervals in a stepped-wedge design. We used latent profile analysis (LPA) to identify oncologist phenotypes based on 11 baseline demographic and practice pattern variables identified using EHR and internal administrative sources. Difference-in-differences analyses assessed associations between oncologist phenotype and the outcome of change in ACP conversation rate, before and during the intervention period. Primary analyses were adjusted for patients' sex, age, race, insurance status, marital status, and Charlson comorbidity index.

**Funding:** This study was funded by the National Palliative Care Research Center (http://www.npcrc.org/) Kornfeld Scholars Award (to RBP). NCI K08CA263541. The sponsors played no role in the study design, data collection and analysis, decision to publish, or preparation of the manuscript.

**Competing interests:** This study was funded by a grant from the National Palliative Care Research Center. The authors have declared that no competing interests exist.

The sample consisted of 2695 patients with a mean age of 64.9 years, of whom 72% were White, 20% were Black, and 52% were male. 78 oncology clinicians (42 oncologists, 36 advanced practice providers) were included. Three oncologist phenotypes were identified: Class 1 (n = 9) composed primarily of high-volume generalist oncologists, Class 2 (n = 5) comprised primarily of low-volume specialist oncologists; and 3) Class 3 (n = 28), composed primarily of high-volume specialist oncologists. Compared with class 1 and class 3, class 2 had lower mean clinic days per week (1.6 vs 2.5 [class 3] vs 4.4 [class 1]) a higher percentage of new patients per week (35% vs 21% vs 18%), higher baseline ACP rates (3.9% vs 1.6% vs 0.8%), and lower baseline rates of chemotherapy within 14 days of death (1.4% vs 6.5% vs 7.1%). Overall, ACP rates were 3.6% in the pre-intervention wedges and 15.2% in intervention wedges (11.6 percentage-point difference). Compared to class 3, oncologists in class 1 (adjusted percentage-point difference-in-differences 3.6, 95% CI 1.0 to 6.1, p = 0.006) and class 2 (adjusted percentage-point difference-in-differences 12.3, 95% confidence interval [CI] 4.3 to 20.3, p = 0.003) had greater response to the intervention.

## Conclusions

Patient volume and time availability may be associated with oncologists' response to interventions to increase ACP. Future interventions to prompt ACP should prioritize making time available for such conversations between oncologists and their patients.

## Introduction

End-of-life care is often not concordant with the goals and wishes of patients with cancer [1]. Early advance care planning has been shown to improve goal-concordant care, decrease end-of-life spending, decrease aggressive care in cancer, and improve patient mood [2–4]. Advances in machine learning (ML) may enable better identification of patients at the highest risk for mortality in order to target interventions for earlier advance care planning discussions (ACPs) [5–10].

Several studies have demonstrated promise in increasing guideline-concordant practice through behavioral interventions targeted towards clinicians [11,12], and there has been similar interest in leveraging behavioral principles to increase the frequency of advance ACP between oncologists and patients. Previous work suggests that targeted ML-based interventions to clinicians can dramatically increase ACPs and palliative care utilization among patients with serious illness. One pragmatic randomized control trial found that an ML-based prompt to oncology clinicians increased rates of ACPs from 3% to 15% of all patients at a large academic cancer center [5,6]. Similar ML-based interventions have been shown to increase ACP documentation [13], reduce length of stay, and increase home palliative care referrals [14]. However, clinicians have heterogeneous responses to such strategies [11], and the efficacy of such interventions across oncology clinician subgroups is not well understood. Identifying subgroups of oncology clinicians that may be more inclined to respond to behavioral interventions to improve ACP may increase the overall effectiveness of such interventions.

Latent profile analysis (LPA) is a hypothesis-free statistical approach to identification of clusters of clinicians based on input variables, and has been used in prior studies to identify phenotypes of patients based on a variety of input data types including clinical [15,16], behavioral [17–19], and activity data [17,20,21]. LPA based on clinician demographics and practice

patterns may help identify groups of clinicians with differing engagement and response to behavioral interventions to improve ACP frequency. In this secondary analysis of a randomized trial, we derived oncologist phenotypes using LPA and compared ACP rates before and after the intervention by phenotype. We hypothesized that distinct clusters of clinicians would be identified by LPA, with variation in response to the ML-based intervention tested in the trial across clusters of clinicians. Our findings provide an empirical approach to phenotype response to ML interventions in healthcare in order to refine such interventions.

## Methods

The University of Pennsylvania Institutional Review Board approved the study. A waiver of informed consent was granted because this was an evaluation of a health system initiative that posed minimal risk to clinicians and patients.

### Study design

This was a secondary analysis of a stepped-wedge randomized trial conducted between June 17 to November 1, 2019 which showed that ML-based nudges among 42 specialty or general oncologists, many of whom worked with an advanced practice provider (APP) as an oncologist-APP dyad (78 total clinicians) caring for 14,607 patients led to a quadrupling of ACP rates (NCT03984773). Eligible clinicians in this secondary analysis included physicians and APPs (physician assistants and nurse practitioners) at 9 medical oncology practices within a large tertiary academic center that participated in the trial. We chose oncologist-APP dyads as the unit of analysis because oncologists usually work 1:1 with APPs in our practice and because oncologists and APPs share responsibility for ACPs for patients. Patients of participating oncologists were excluded if they had a documented ACP prior to the start of the trial, or if they were enrolled in another ongoing trial of early palliative care. Medical genetics encounters were also excluded.

### Outcome

The primary outcome was the change in ACP rate among all encounters with patients with >10% predicted 180-day mortality risk in the intervention period compared to the pre-intervention period. Any note which utilized the ACP template in the electronic medical record was classified as an ACP.

### Intervention

The clinical trial used an ML algorithm which generated predictions of 180-day mortality for cancer patients, and a multi-pronged behavioral intervention to increase ACP frequency based on the generated predictions. The ML algorithm incorporated 3 classes of variables 1) demographic variables, 2) Elixhauser comorbidities and 3) laboratory and select electrocardiogram data. The algorithm utilized a gradient boosted algorithm to identify patients at risk of short-term mortality [22]. Clinicians caring for patients at high risk of short-term mortality predicted risk of mortality >10% were prompted to initiate an ACP through a multipronged intervention incorporating principles of behavioral economics including peer comparisons, performance reports, and opt-out default text messages based on the ML algorithm. Because clinicians received the intervention only for patients with >10% predicted risk of mortality, our primary analysis only included patients with >10% predicted risk of mortality in order to restrict our cohort to the target population of the intervention. Further details of the intervention and clinical trial are published elsewhere [5,6].

## Data

11 variables were included in this study based on their conceptual relevance to a clinician's expected response to the ML intervention. The selected variables were grouped into three categories: demographic, practice pattern, and end-of-life outcomes. Demographic variables included the clinician's gender and years in practice. Practice pattern variables included the clinician's oncology subspecialty (e.g. general oncology, thoracic, genitourinary, etc.); number of days in clinic per week [1–5]; percentage of patient encounters with new patients (0–100%); average number of patient encounters per week (continuous); average number of encounters per day; number of years in practice; and baseline ACP rates in the month prior to the start of our randomized trial. End-of-life outcomes metrics were measured in the year prior to the start of our trial among patients who died and who were part of an oncology clinician's panel. These variables included chemotherapy received within 14 days of death, death in the hospital, and hospice enrollment prior to death. Practice pattern and end-of-life outcome data came was obtained from Clarity, an EPIC reporting database that contains structured data elements of individual EHR data for patients treated at the University of Pennsylvania Health System. Demographic data and years in practice were extracted from an internal database of the Abramson Cancer Center at Penn Medicine.

## Oncologist phenotyping

We used latent profile analysis (LPA), applied to the aforementioned variables, to identify phenotypes of oncologists based on their demographic information and practice patterns. LPA is a statistical modelling approach for recovering hidden groups in data by modeling the probability that individuals in the dataset belong to different groups [23]. LPA is conceptually similar to Latent Class Analysis, however, LPA enables recovery of hidden groups based on continuous data whereas latent class analysis is only suitable for analysis of categorical data. Since most of the variables chosen in our analysis are continuous, we used LPA instead of latent class analysis. 11 variables described in the data section were included in the LPA. These variables were not standardized in the analysis as it has no impact on the results of the clustering algorithm. To determine the model of best fit, we used the Akaike information criterion (AIC), Bayesian information criterion (BIC), and entropy. AIC and BIC are two estimators of a model's prediction error which balance the goodness of fit with model simplicity [24]. Entropy is a commonly used statistical measure of the separation between classes in LPA [25]. The Bootstrapped Likelihood Ratio Test (BLRT) was also used to assess whether a given model with k classes is significantly more informative than one with k-1 classes [26]. We required that each class contain a minimum of 10% of oncologists (n = 5 oncologists). LPA was conducted using the tidyLPA package in R version 3.6.0 [27]. We attached descriptive labels to each of the clusters in order to provide interpretability to the clustering results. Means were calculated and examined for each of the 11 variables included in the clustering analysis, and labels were selected to capture clinically relevant themes shared by most of the clinicians in the cluster, and to capture variability between clusters.

## Statistical analysis

Difference-in-difference analyses tested the association between the identified oncologist phenotypes and response to the nudge. Changes in the ACP rate (pre-intervention vs. intervention period) were compared for each phenotype identified by LPA. We fit a multivariable logistic regression model using the clinician phenotype as a predictor for whether the patient received an ACP or not at the patient-level. Covariates included in the model were the interaction term between oncologist phenotype and intervention period, patients' age (continuous), gender,

race, insurance type, marital status, and Charlson comorbidity score. Adjusted probabilities of receiving an ACP accounted for these variables and were calculated by converting the log-odds ratio from the model output for each class pre-intervention and in the intervention period into a probability. Difference-in-difference estimates comparing class 1 and class 2 to class 3 were calculated by taking the difference in intervention response as measured by the difference in pre-intervention adjusted probability of ACP and intervention period adjusted probability of ACP for each of the classes. The adjusted probabilities and difference-in-difference in percentage points with 95% confidence intervals were estimated by bootstrapping, where the data was resampled 1000 times. Statistical significance of the difference-in-differences was calculated by the p-value of the interaction between clinician phenotype and intervention.

In a secondary analysis, we used logistic regression to measure the impact of various clinician-level variables on the likelihood of a patient receiving an SIC in both the pre-intervention and intervention periods. The logistic regression was conducted at the level of the patient-wedge with the outcome of SIC receipt. Patient covariates included in the model were patient sex, age, race, insurance status, marital status, and Charlson Comorbidity Index. Clinician-level variables included in model were the number of days in clinic per week, percentage of new patients per week, average patients per week, average encounters per day, years in practice, and end-of-life quality metrics (hospice enrollment rate, inpatient death rate, and chemo utilization at the end of life). All analyses were conducted using R version 3.6.0.

### Sensitivity analysis

To analyze whether response to the intervention was similar among all patients regardless of predicted risk of mortality, we applied the aforementioned analysis to all patients, including those with predicted risk of mortality of less than 10%. We compared response to the ML-based intervention by clinician phenotype identified by LPA as described above in Statistical Analysis.

## Results

The trial sample consisted of 78 clinicians (of whom 42 were oncologists), 14 607 patients, and 26 059 patient encounters (**Fig 1**). In this secondary analysis of a pragmatic randomized control trial, we studied a subset of oncologists and their patient encounters that included ACPs.

### Clinician characteristics

We studied 42 oncologist and oncologist-APP dyads in this analysis. Among oncologists, 26 (61.9%) were male and 16 (38.1%) were female. 6 (14.3%) were general oncologists and 36 (86%) were specialty oncologists. The median number of years in practice was 7.4 (IQR 5.3, 13.0), and oncologists spent a mean of 2.8 SD (1.1) days in the clinic per week and saw an average of 28.7 SD (15.2) patients per week. The median percentage of new patients seen per week was 21% (IQR 15.8%, 24.1%), and median number of encounters per day was 9.3 (IQR 8.0,11.5).

### Model selection

Models with two latent classes and three latent classes were generated. The entropy of the 2-class model and 3-class model were comparable. The 3-class model was selected as the model of best fit by the BLRT (p = 0.010) and because 3-class model had a lower AIC (2678.46 vs. 2689.46) (**S1 Table**). In addition, this model was reviewed by the first and senior authors

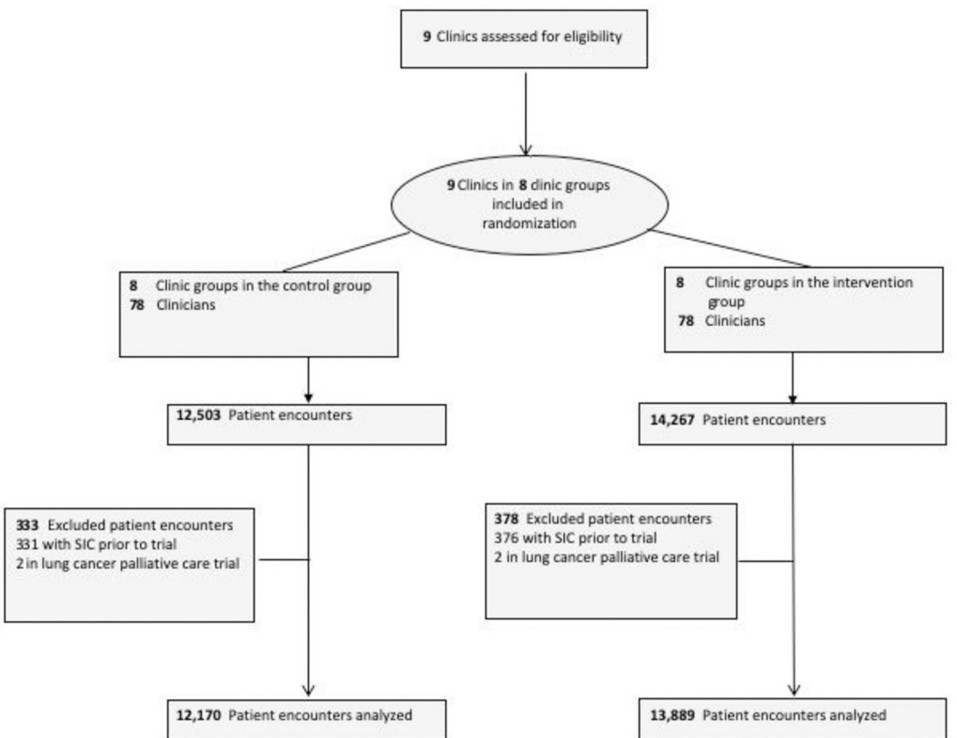

**Fig 1. CONSORT diagram.** SIC indicates serious illness conversation, a type of ACP.

for clinical interpretability and chosen because the 3-class model distinguished between high and low volume specialty clinicians. This model was chosen to ensure the model did not collapse potentially meaningfully different classes into a single class given comparable statistical estimates of prediction error between the 2-class and 3-class models. Each of the three latent classes contained greater than 10% of the total clinician population. Based on this model, three oncologist phenotypes were identified (**Table 1**).

**Class 1.** This class was comprised of 9 oncologists, containing 21% of the total clinician population. Of the three classes, these oncologists had the most years in practice (mean [range], 8.42 [3.59, 37.0]), saw the most patients per week (mean standard deviation, SD): 53.2 [8.9]), had the highest number of clinic days per week (mean [SD]: 4.4 [0.7]), had the lowest percentage of new patients per week (mean [SD]: 17% [5.7%]) and had lowest baseline ACP rates (mean [SD]: 0.8% [0.7%]), highest chemotherapy use rates within 14 days of death (mean [SD] 7.1% [7.5%]) and intermediate inpatient death rates(mean [SD] 9.9% [6.9%]). This class is comprised primarily of generalist oncologists with high-volume practices.

**Class 2.** This class was comprised of 5 specialty oncologists, containing 12% of the total study population. Of the three classes, this class had the fewest years in practice (mean [range] 5.26 [2.39, 21.0]), saw the fewest patients per week (mean [SD]: 9.2 [5.6]), had the fewest clinic days per week (mean [SD]: 1.6 [0.9]), saw the highest percentage of new patients per week (mean [SD]: 34% [13.1%]), had the highest baseline ACP rates (mean [SD]: 3.9% [5.0%]), lowest chemotherapy use rates within 14 days of death (mean [SD]: 1.4% [2.8%]) and lowest inpatient death rates (mean [SD]: 5.8% [4.3%]). This class is comprised primarily of specialist oncologists with low-volume practices.

**Class 3.** This class was the largest class, comprised of 28 specialty oncologists containing 67% of the study sample. Of the three classes, this class tended to have an intermediate number

**Table 1. Demographic and practice characteristics of oncologist phenotypes.**

| | Class 1 (N = 9) | Class 2 (N = 5) | Class 3 (N = 28) | Overall (N = 42) |
|---|---|---|---|---|
| **Gender** | | | | |
| Female | 3 (33.3%) | 2 (40.0%) | 11 (39.3%) | 16 (38.1%) |
| Male | 6 (66.7%) | 3 (60.0%) | 17 (60.7%) | 26 (61.9%) |
| **Practice specialty** | | | | |
| General Oncology | 6 (66.7%) | 0 (0%) | 0 (0%) | 6 (14.3%) |
| Specialty Oncology | 3 (33%) | 5 (100%) | 28 (100%) | 36 (82%) |
| **Years in practice** | | | | |
| Median [Min, Max] | 8.42 [3.59, 37.0] | 5.26 [2.39, 21.0] | 7.43 [2.06, 31.5] | 7.43 [2.06, 37.0] |
| **Days in Clinic Per Week** | | | | |
| Mean (SD) | 4.44 (0.726) | 1.60 (0.894) | 2.50 (0.577) | 2.81 (1.11) |
| **Percentage of New Patients Per Week** | | | | |
| Median [Min, Max] | 21% [6%, 25%] | 32% [24%,52%] | 21% [11%, 36%] | 21% [6%, 52%] |
| Missing | 0 (0%) | 1 (20.0%) | 0 (0%) | 1 (2.4%) |
| **Average number of patients per week** | | | | |
| Median [Min, Max] | 50.2 [39.8, 65.7] | 10.5 [0.900, 15.5] | 24.9 [13.6, 36.4] | 25.4 [0.900, 65.7] |
| **Average number of encounters per day** | | | | |
| Median [Min, Max] | 12.3 [10.3, 14.4] | 5.65 [1.83, 9.88] | 9.15 [6.33, 15.5] | 9.33 [1.83, 15.5] |
| **Baseline ACP rate** | | | | |
| Mean (SD) | 0.801% (0.666) | 3.94% (4.95) | 1.59% (1.50) | 1.70% (2.18) |
| **Hospice Enrollment rate at baseline** | | | | |
| Mean (SD) | 69.4% (17.9) | 59.2% (10.7) | 61.3% (28.4) | 62.9% (25.0) |
| Missing | 0 (0%) | 1 (20.0%) | 1 (3.6%) | 2 (4.8%) |
| **Inpatient death rate at baseline** | | | | |
| Mean (SD) | 9.88% (6.89) | 5.81% (4.28) | 17.2% (11.1) | 14.4% (10.5) |
| Missing | 0 (0%) | 1 (20.0%) | 1 (3.6%) | 2 (4.8%) |
| **Chemotherapy use at end of life at baseline** | | | | |
| Mean (SD) | 7.06% (7.46) | 1.39% (2.78) | 6.50% (6.95) | 6.11% (6.84) |
| Missing | 0 (0%) | 1 (20.0%) | 1 (3.6%) | 2 (4.8%) |

of years in practice (mean [range] 7.43 [2.06, 31.5]), saw an intermediate number of patients per week (mean [SD]: 24.3 [5.8]), had an intermediate number of clinic days per week (mean [SD]: 2.5 [0.6]), intermediate percentage of new patients per week (mean [SD]: 21% [6.2%]) and had an intermediate baseline ACP rates (mean [SD]: 1.6% [1.5%]) as well as highest inpatient death rates(mean [SD]: 17.2% [11.1%]), and intermediate rates of chemotherapy use within 14 days of death (mean [SD]: 6.5% [7.0%]). This class is comprised primarily of specialist oncologists with high-volume practices.

## Intervention response by clinician phenotype for high-risk patients

The probability of a high-risk patient (predicted 180-day mortality >10%) receiving an ACP increased significantly following the intervention among patients receiving care from class 1 and class 2 oncologists compared to class 3 oncologists. Among patients receiving care from class 3 oncologists, the adjusted probability of a high-risk patient receiving an ACP increased from 2.3% pre-intervention to 7.6% during the intervention period. Among patients receiving care from class 2 oncologists, the adjusted probability of ACP increased from 3.1% pre-intervention to 20.7% in the intervention period (adjusted percentage-point difference-in-differences relative to class 3 oncologists 12.3, 95% CI 4.3 to 20.3, p = 0.003) (**Table 2**). Class 1

**Table 2. Association between oncologist phenotype and response to the intervention.**

| Phenotype | Oncologists, n (%) | Patients, n (%) | Adjusted probability of ACP, pre-intervention | Adjusted probability of ACP, intervention | Percentage point difference in differences vs Class 3 (95% CI) | p-value |
|---|---|---|---|---|---|---|
| Class 3 | 28 (67%) | 1883 (70%) | 2.3% | 7.6% | — | — |
| Class 1 | 9 (21%) | 673 (25%) | 1.9% | 10.7% | 3.6 (1.0,6.1) | 0.006 |
| Class 2 | 5 (12%) | 139 (5%) | 3.1% | 20.7% | 12.3 (4.3, 20.3) | 0.003 |

oncologists also had a significantly greater response relative to class 3 oncologists (adjusted percentage-point difference-in-differences relative to class 3 oncologists 3.6, 95% CI 1.0, 6.1, p = 0.006), though the magnitude of this change was not as large as that of class 2 oncologists. The adjusted probability of ACP for class 1 oncologists increased from 1.9% pre-intervention to 10.7% in the intervention period (**Fig 2**).

Multivariable logistic regression models were run at the patient level for patients with a predicted 180-day mortality risk of greater than 10% using the clinician phenotype as a predictor for whether the patient received an ACP or not. Covariates included in the model included the interaction term between oncologist phenotype and intervention period, patients' age

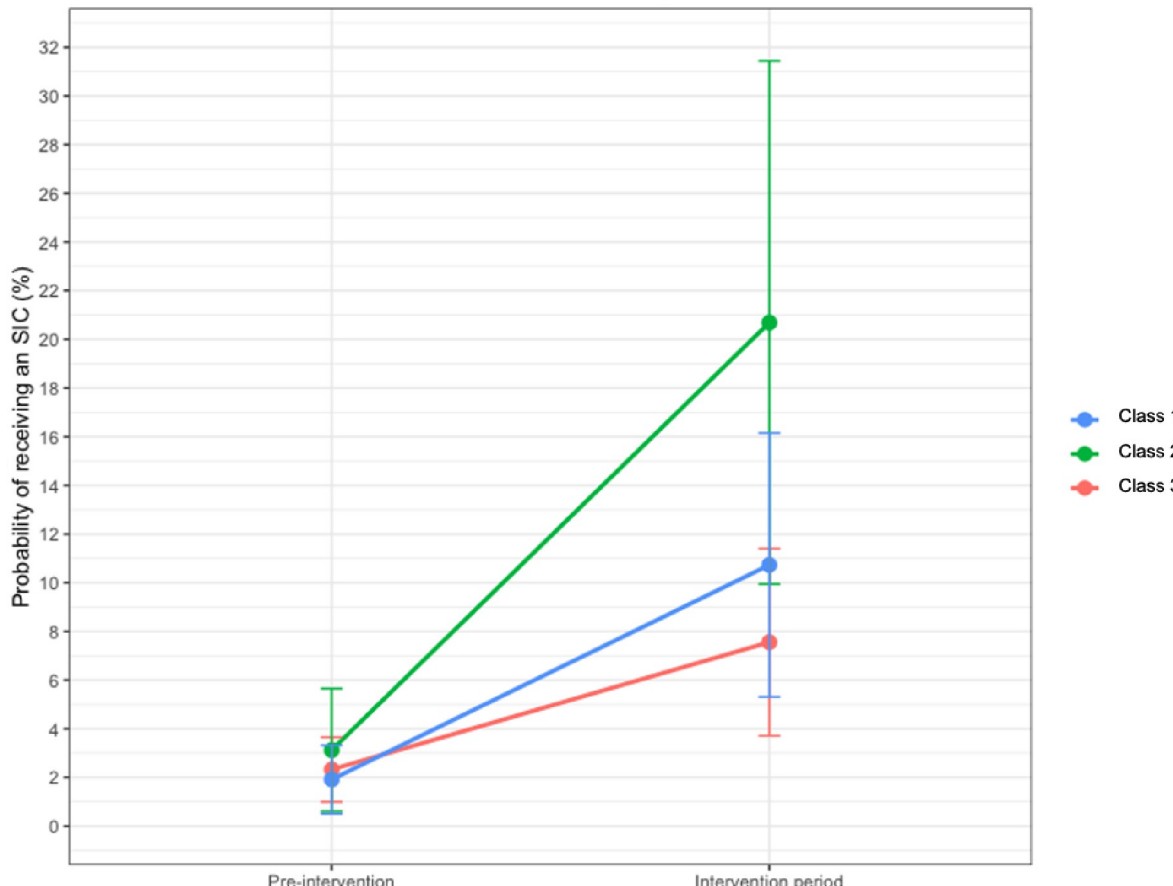

**Fig 2. Intervention response by oncologist phenotype for patients with high predicted risk of mortality.** The adjusted probability of a high risk patient (predicted 180-day mortality risk >10%) of receiving an SIC during the pre-intervention and intervention periods by oncologist phenotype. Class 2 oncologists (green) had the highest response to the intervention, with the probability of receiving an ACP increasing from 3.1% during the pre-intervention period to 20.7% during the intervention period. The adjusted probability of ACP increased from 1.9% to 10.7% among class 1 oncologists (blue), and from 2.3% to 7.6% for class 3 oncologists (red).

(continuous), gender, race, Insurance type, marital status, and Charlson comorbidity score. Adjusted probabilities of receiving an ACP accounted for these variables and were calculated by converting the log-odds ratio from the model output for each oncologist class pre-intervention and during the intervention period into a probability. The adjusted probabilities and difference-in-difference in percentage points with 95% confidence intervals were estimated by bootstrapping, where the data was resampled 1000 times.

### Sensitivity analyses: Intervention response by clinician phenotype for all patients in the study cohort

As a sensitivity analysis, we compared the probability of ACP before and after the intervention for all patients (not only high risk patients) across clinician phenotypes. Consistent with the main analysis, the probability of ACP for all patients increased significantly more for class 2 oncologists compared to class 3 oncologists (adjusted percentage-point difference-in-difference 2.6, 95% CI 0.9 to 4.3, p = 0.002). (**S2 Table**) The change in ACP rate was not statistically significant for class 1 oncologists compared to class 3 oncologists (adjusted percentage-point difference-in-difference 0.2, 95% CI 0 to 0.4, p = 0.109) (**S1 Fig**).

Multivariable logistic regression models were run at the patient level for all patients in the cohort using the clinician phenotype as a predictor for whether the patient received an ACP or not. Covariates included in the model included the interaction term between oncologist phenotype and intervention period, patients' age (continuous), gender, race, Insurance type, marital status, and Charlson comorbidity score. Adjusted probabilities of receiving an ACP accounted for these variables and were calculated by converting the log-odds ratio from the model output for each oncologist class pre- and post-intervention into a probability. The adjusted probabilities and difference-in-difference in percentage points with 95% confidence intervals were estimated by bootstrapping, where the data was resampled 1000 times.

### Logistic regression on oncologist characteristics associated with likelihood of SIC

In our adjusted secondary regression analysis, specialist oncologists, higher number of days per week in clinic, and higher percentage of new patients per week were associated with significantly greater likelihood of SIC receipt (**S3 Table**).

### Discussion

In this secondary analysis of a randomized trial analyzing oncology clinician response to an ML-based intervention to increase ACP frequency, we identified three phenotypes of oncology clinicians based on demographic, practice pattern, and end-of-life quality data. While our overall trial was associated with an 11.6 percentage-point increase in ACPs, we found that this response varied considerably among each of the 3 identified phenotypes. In particular, the intervention was associated with a 5.6-fold and 6.7-fold increase in response rates among class 1 oncologists, who consisted primarily of general oncologists with higher patient volumes; and class 2 oncologists, who consisted primarily of specialists with lower patient volumes; compared to class 3 oncologists, who consisted primarily of specialists with higher patient volumes. While prior studies have identified groups of clinicians who vary in their surveyed attitudes towards ML-based clinical support tools [28], this is one of the first studies to identify phenotypes of clinician response to an ML-based clinical intervention studied in a randomized controlled trial and demonstrate significant variation in response to the intervention by phenotype. These findings are consistent with prior analyses, which have demonstrated the

feasibility of using a variety of data sources including clinical [15,16], behavioral [17–19], and activity data [17,20,21] to identify subgroups of clinicians and patients with different responses to interventions. These findings have several important implications for future design of ML interventions, particularly those to improve care of advanced illness.

First, this analysis suggests mechanisms by which ML-based interventions may increase advance care planning in previous trials [6,13,14]. One possible reason for variable response to an ML-based intervention observed in this study is variation in cognitive workload. Prior studies of physician behavior have found that the frequency of desired behaviors requiring active cognitive effort such as influenza vaccination, antibiotic prescribing, and hand hygiene decline over the course of the day as cognitive workload builds [29–31]. Class 2 oncologists may have responded more strongly to this ML-based intervention due to several factors, including having more time to spend with their patients due to the lower practice volume. Such clinicians also had better baseline performance of ACPs, suggested by their higher baseline rates of ACPs and higher concordance with clinical practice guidelines for end-of-life care. While this analysis did not exhaustively examine all provider and practice pattern characteristics of these oncology clinicians, our analysis suggests that bandwidth and patient volume may be drivers of response to interventions intended to improve advance care planning and clinician-patient interaction.

Second, this analysis offers insights into targeting ML-based interventions. Our analysis argues to focus ML-based interventions on clinician phenotypes that may be more likely to respond to such interventions. In contrast, clinicians and health systems should pay careful attention to resource constraints before deploying potentially expensive ML interventions to clinicians with higher patient volumes, who may be less likely or able to respond. While ML-based interventions or EHR-based clinical decision support usually pose little risk to patient safety and outcomes, some studies have found evidence of "alert fatigue" [32] among clinicians. As our present study demonstrates, a small cluster of clinicians may respond strongly to a particular intervention while most clinicians exhibit less response, limiting broad application of the intervention to all clinicians in a practice setting. Targeted deployment of ML-based interventions in the future to clinicians most likely or able to respond, while mitigating alert fatigue or workflow interruptions for clinicians less likely to respond, is a viable strategy for future deployment of ML-based clinician decision support tools.

Third, while techniques to characterize patient phenotypes have been utilized in population health to identify patients for targeted interventions for behavior change [33,34], the application of similar techniques to identify groups of clinicians with differential response to ML-based interventions is relatively unexplored [11]. Utilizing clinician-level data available in institutional data stores or EHRs may provide additional insights into clinician behavior and enable better understanding of clinician response to future ML-based interventions and health systems initiatives. Using such techniques allows for better description of which clinicians are responding to an intervention and the magnitude of response. Leveraging the availability of EHR and additional sources of clinician-level data, combined with hypothesis-free techniques for identification of hidden clusters within data, may provide a clearer way to interrogate the efficacy and responses to ML-based interventions.

This study has several limitations. First, this trial was conducted within a single tertiary cancer center with limited sample size. The results of our analysis may be influenced by features of individual oncologists who practice at our center, and the results of this study may be difficult to generalize to other settings whose characteristics of oncologists differ from our sample. However, each cluster includes at least 10% of the study population which insulates our results against inappropriate influence of any single clinician on cluster characteristics. Furthermore, our findings regarding the potential association of patient volume with intervention

effectiveness is likely generalizable given the intuitive reasons that lower-volume clinicians likely have more time and clinical bandwidth to have these conversations. Additionally, the study included clinicians who practiced at either academic and/or community sites and includes diverse patients across demographics, socioeconomic, cancer type, and comorbidity domains. Thus, we believe this is generalizable to a large proportion of oncology practices and practicing oncologists.

Second, we were also limited to studying the effect of the intervention on ACP frequency, as we did not have adequate follow-up to determine the effect of the intervention on end-of-life outcomes. However, ACPs are a guideline-based quality metric in cancer and other advanced illnesses and a surrogate for downstream goal-concordant care [35–37]. Future analyses may study the impact of ML interventions on metrics such as inpatient death rates, chemotherapy utilization, and hospice enrollment, and how the impact of ML-based interventions may vary by clinician phenotype.

## Conclusion

Among three phenotypes of oncologists identified by LPA at a large academic medical center, an ML-based intervention to increase ACP frequency had greater effect on class 1 oncologists, which were generally comprised of high-volume generalists, and class 2 oncologists, which were generally comprised of low-volume specialists, compared to class 3 oncologists, which were generally comprised of high-volume specialists. Not all oncologists respond similarly to ML-based interventions, and response to ML-based interventions to guide clinician behavior may in part be determined by a clinician's cognitive workload and patient volume. Future initiatives to prompt ACP conversations between oncology clinicians and patients should prioritize making time available for such conversations, in order to maximize clinician response.

## Supporting information

**S1 Checklist.**
(DOC)

**S1 Fig. Intervention response by oncologist phenotype for all patients in the study cohort.**
The adjusted probability of any patient in the cohort receiving an SIC during the pre-intervention and intervention periods by oncologist phenotype. Class 2 oncologists (green) had the highest response to the intervention, with the probability of receiving an SIC increasing from 0.5% during the pre-intervention period to 3.8% during the intervention period. The adjusted probability of ACP increased from 0.2% to 1.0% among class 1 oncologists, and from 0.3% to 0.9% for class 3 oncologists.
(DOCX)

**S1 Table. Model fit statistics by number of classes included in the model.**
(DOCX)

**S2 Table. Association between oncologist phenotype and response to nudges (whole cohort).**
(DOCX)

**S3 Table. Logistic regression at the patient-wedge level identifying clinician characteristics associated with increased likelihood of conducting an SIC.**
(DOCX)

**S1 File.**
(CSV)

**S2 File.**
(CSV)

## Author Contributions

**Conceptualization:** Eric Li, Christopher Manz, Manqing Liu, Jinbo Chen, Corey Chivers, Jennifer Braun, Lynn Mara Schuchter, Pallavi Kumar, Mitesh S. Patel, Lawrence N. Shulman, Ravi B. Parikh.

**Data curation:** Eric Li, Christopher Manz, Manqing Liu, Jinbo Chen, Corey Chivers, Mitesh S. Patel, Ravi B. Parikh.

**Formal analysis:** Eric Li, Manqing Liu, Jinbo Chen, Ravi B. Parikh.

**Funding acquisition:** Ravi B. Parikh.

**Investigation:** Eric Li, Manqing Liu, Ravi B. Parikh.

**Methodology:** Eric Li, Christopher Manz, Manqing Liu, Jinbo Chen, Corey Chivers, Jennifer Braun, Lynn Mara Schuchter, Pallavi Kumar, Mitesh S. Patel, Lawrence N. Shulman, Ravi B. Parikh.

**Project administration:** Ravi B. Parikh.

**Resources:** Ravi B. Parikh.

**Software:** Eric Li, Manqing Liu, Corey Chivers.

**Supervision:** Jinbo Chen, Ravi B. Parikh.

**Validation:** Eric Li, Christopher Manz, Manqing Liu, Jinbo Chen, Corey Chivers, Jennifer Braun, Lynn Mara Schuchter, Pallavi Kumar, Mitesh S. Patel, Lawrence N. Shulman, Ravi B. Parikh.

**Visualization:** Eric Li, Manqing Liu, Ravi B. Parikh.

**Writing – original draft:** Eric Li.

**Writing – review & editing:** Eric Li, Christopher Manz, Manqing Liu, Jinbo Chen, Corey Chivers, Jennifer Braun, Lynn Mara Schuchter, Pallavi Kumar, Mitesh S. Patel, Lawrence N. Shulman, Ravi B. Parikh.

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
