## [Decision Letter · Decision Letter 0]

26 Oct 2021

PONE-D-21-24615Oncologist Phenotypes and Associations with Response to a Machine Learning-Based Intervention to Increase Advance Care Planning: Secondary Analysis of a Randomized Clinical TrialPLOS ONE

Dear Dr. Parikh,

Thank you for submitting your manuscript to PLOS ONE. After careful consideration, we feel that it has merit but does not fully meet PLOS ONE’s publication criteria as it currently stands. Therefore, we invite you to submit a revised version of the manuscript that addresses the points raised during the review process.

I want to apologize for the length of time it took to review this manuscript. As you can see we obtained reviews from 3 independent reviewers including one biostatistician. The overall tenor of the reviews is positive and I believe the comments/suggestions provided are reasonable.

We look forward to receiving your revised manuscript.

Kind regards,

Randall J. Kimple

Academic Editor

PLOS ONE

Journal Requirements:

Reviewers' comments:

Reviewer's Responses to Questions

**Comments to the Author**

1. Is the manuscript technically sound, and do the data support the conclusions?

Reviewer #1: Partly

Reviewer #2: Partly

Reviewer #3: Yes

2. Has the statistical analysis been performed appropriately and rigorously? 

Reviewer #1: I Don't Know

Reviewer #2: No

Reviewer #3: Yes

3. Have the authors made all data underlying the findings in their manuscript fully available?

Reviewer #1: No

Reviewer #2: No

Reviewer #3: No

4. Is the manuscript presented in an intelligible fashion and written in standard English?

Reviewer #1: Yes

Reviewer #2: Yes

Reviewer #3: Yes

5. Review Comments to the Author

Reviewer #1: In this secondary analysis of a randomized clinical trial that tested an intervention to increase ACP at a tertiary cancer hospital, the authors performed an analysis of oncologist phenotype to assess whether that impacted the likelihood of increasing ACP discussions.

I have 1 major issue with the article and several minor ones below. Mostly, it is very unclear how the phenotypes of the physicians are actually defined. For example, why were there some specialists listed in the generalist phenotype and it is unclear how low volume specialists versus high volume specialists were exactly defined. This makes it very difficult to truly understand the difference between the groups.

Minor issues:

1) Why was the initial analysis only done on patient with >10% risk of mortality, with sensitivity analysis on the entire cohort? Why not just do the initial analysis on the entire cohort.

2) In the abstract (page 2 line 36) please add the word patients' when explaining the multivariable analysis variables.

3) When explaining the statistics behind the oncologist phenotyping please explain why you decided to focus on patient volume rather than other provider characteristics (main center versus satellite, age, number of years in practice, etc.) Seems like the authors did have an underlying hypothesis when choosing patient volume as the phenotype to test.

4) Please clarify if the 78 clinicians studied were only in the intervention group of the randomized trial, and if so, shouldn't there only have been 12,170 patient encounters? If both groups were analyzed shouldn't there have been more physicians?

5) As already stated above in the major issues, why were there 3 specialty oncology physicians in the generalists phenotype (how the phenotypes were defined needs to be made more clear).

6) The wording in the response section when describing the difference in response rates between the specialists needs to be a more little more careful. The authors were not studying the individual doctor response rates. Rather they were testing whether the patients were listed as having an SIC and comparing that between the doctor phenotypes. It may seem irrelevant, but it is important statistically as it implies that the individual doctors were being tested for how well they responded to the intervention which is not the case. The statistics were done on a patient level.

7) Discussion: "our analysis suggests that bandwidth and volume are key drivers of response to interventions intended to improve advance care planning and clinician-patient interaction." Please be careful here with your wording. Your analysis does not suggest that these items are key drivers, rather it suggests that they "may be a driver of". Youre multivariate model only included one phenotypic characteristic of physicians (all of the other variables were patient variables). Therefore, you have no idea whether patient volume or other provider variables may be the actual driver. Maybe patient volume is associated with another physician variable (like training programs, gender, etc.) that are the actual drivers. This analysis did not look at that and therefore cannot claim this one provider variable is the key driver.

8) Discussion: "Targeted deployment of ML-based interventions in the future to clinicians most likely or able to respond, while mitigating alert fatigue or workflow interruptions for clinicians less likely to respond, is a viable strategy for future deployment of ML-based clinician decision support tools." One could also argue that an intervention that cannot be implemented by the physicians that are seeing the majority of the patients is probably not a good intervention. If only the doctors who are seeing the least number of patients can intervene than the majority of patients will not be helped by the intervention.

Reviewer #2: Thank you to the authors for their hard work and submission and for the opportunity to review this study.

This is an interesting secondary analysis of a recently reported trial investigating the use of machine learning to direct behavioral nudges for advanced care planning discussions. This study explores practitioner characteristics to identify potential groups where the intervention may have had a greater effect on practice. Overall, the study is important, a good idea, and interesting. The investigators should be applauded for their work in this area, though I do have a number of comments for clarification, particularly around study design and the conclusions drawn by the authors.

Major comments:

1) Overall comment: the use of LPA is creative to try to define clusters/groups of physicians that respond differently to the intervention. However, this is overall less interpretable and more complex, which is reflected in the discussion. Overall, it seems like the investigators' primary objective is to identify characteristics associated with response. To that end, a logistic regression model across characteristics may be the most helpful tool, and I believe it should be included in the study even if it does not end up as a point of emphasis. Otherwise, conclusions are discussed in the context of clusters whose names are potentially overly simplified (comments regarding this challenge below). Rather than generating logistic regression models summarizing physician features with the LPA, it may be more clear to do so with physician characteristics themselves. It may also reduce some of the challenges with small categories caused by the LPA approach. The overall advantage of using the less-transparent LPA approach feels a bit unclear (and less practically useful).

2) LPA: I have a couple of questions for clarification - continuous data is on multiple scales (in this study for instance, clinic days/week versus % new patients versus patient encounters/week). Were these data standardized? Were baseline ACP rates ascertainable from Clarity?

3) The authors used AIC/BIC/entropy approaches to determine the best fit model. More on this decision making process should be discussed (balancing AIC/BIC, etc). AIC/BIC approaches do also have limitations that have been well-discussed in the statistical literature. The concept of "clinical interpretability" should also be discussed further. It is possible that due to the small sample sizes - particularly in the distribution of some characteristics (which result in imbalanced classes) may not allow the generation of highly distinct classes and that the 1 or 2 class models may not be as overfit as the 3 class model.

4) While the results appear to make sense, I think the authors should discuss the limitations of small sample sizes more. Using only 5 oncologists to define a group limits its external generalizability. Only 6 oncologists in the study were generalists, and only 5 were classified into the "low-volume specialists group." While the overall diversity of the general trial (as the authors have highlighted in the discussion) are an overall credit, this also reduces the sizes of each group and makes it more challenging to characterize the subgroups (increasing the brittleness of each group and potential bias). For instance, conclusions drawn on those 6 generalists is highly dependent on those few oncologists; they would also be expected to cluster together as a small group among specialists. Conclusions drawn here may not reflect differences that would be detected if this study was performed exclusively among generalists for instance. I think this limitation may contribute to comment #2 with regards to the models that had fewer classes.

5) On a related note, the "high-volume generalists" category feels like it may be a misnomer - all 6 of the generalists are in this group (which would be expected as a small minority group), but specialists still make up a fair number in the group. Similarly, the small "low-volume specialists" group also has a particularly high baseline ACP rate and fewer years in practice. Limiting the names to specific dimensions loses the resolution/benefit of including all of the variables in the LPA process. Again, logistic regression would help distill some of these features out (such as volume-based metrics).

6) I think there may be an error in line 197 - I think that 82% may be miscalculated.

7) Data availability - I agree that the patient data may not be available, though with 42 oncologists analyzed, I feel like deidentified individual level data should be potentially available for sharing, and I encourage the authors to consider exploring this given the PLOS Data policy.

Minor comments:

- I think that the "number of oncology clinicians" in Table 1 is a bit misleading as the analysis is based on dyads rather than individual clinicians.

- While the authors indicate primacy in line 327-329 in the discussion, I don't believe this is necessarily true. There have been a number of studies now investigating clinician trust and use of AI, particularly in the radiology space. Some of those findings have similar findings as in this study - for instance more junior/trainee radiologists are more likely to follow clinical decision support tools/computer aided diagnosis systems. This historical work should be included and placed in context.

Reviewer #3: A secondary analysis of a clinical trial aimed to evaluate associations between phenotypes and response to machine learning based intervention to prompt earlier advance care planning for cancer patients. High-volume specialists had greater response to intervention when compared to low-volume specialists and high-volume generalists.

Minor revisions:

1- Line 197: Provide a measure of dispersion, perhaps interquartile range or range, for the median number of years in practice.

2- Line 198: Provide standard deviations for days oncologist spent in clinic and patients seen per week.

3- Line 199-200: Provide a measure of dispersion for these medians.

6. PLOS authors have the option to publish the peer review history of their article (what does this mean?). If published, this will include your full peer review and any attached files.

Reviewer #1: No

Reviewer #2: No

Reviewer #3: No

---

## [Author Response · Author response to Decision Letter 0]

13 Dec 2021

Please see attached response letter, which is pasted below. 

December 11th, 2021

Randall J. Kimple MD, PhD

Academic Editor

PLOS ONE

Dear Dr. Kimple and Reviewers, 

Thank you for your thoughtful review of our manuscript, “Oncologist Phenotypes and Associations with Response to a Machine Learning-Based Intervention to Increase Advance Care Planning: Secondary Analysis of a Randomized Clinical Trial.” We appreciate your comments and have responded to each of your concerns below. Manuscript revisions are highlighted in bold with page numbers indicating pages in the clean, revised version of the manuscript. 

REVIEWER 1 SPECIFIC COMMENTS 

1. It is very unclear how the phenotypes of the physicians are actually defined. For example, why were there some specialists listed in the generalist phenotype and it is unclear how low volume specialists versus high volume specialists were exactly defined. This makes it very difficult to truly understand the difference between the groups.

We thank the reviewer for these thoughtful comments and the opportunity to further clarify the rationale and methodology behind this study. 

We labelled clusters identified by latent profile analysis by calculating means of each of the 11 covariates used in the clustering analysis. To enforce some level of interpretability to the clusters, we then examined these means and assigned descriptive labels to the clusters based on what we perceived to be clinically relevant themes shared by most of the clinicians in the cluster. We now better clarify this in our Methods section. For example, all clusters varied significantly in the average patients seen per week: The mean number of patients seen per week was 24.9, 10.5, and 50.2 among the three clusters. Thus, we labelled each cluster using patient volume was chosen as a covariate to characterize these clusters. Specifically, the label “high-volume” was then given to the first and third clusters because their mean number of patients seen per week was significantly higher in those clusters than in the “low-volume” second cluster. 

As a result, in this analysis, certain generalists were grouped together with specialists because of underlying similarity in other sociodemographic or practice pattern characteristics. Labels such as “generalist” or “specialist” were attached to the clusters based on overarching similarities between clinicians within the cluster and were provided to add clinical interpretability to the results of the analysis. While not every clinician within the cluster may fit the exact labels attached to the cluster, we believe that the labels identify general patterns of similarity between clinicians within the cluster that are clinically relevant. 

In the second section of our analysis, each clinician cluster had meaningful differences in response to the intervention, reinforcing the distinctiveness of the clusters identified in this analysis.

We have clarified our process for generating cluster labels in our revised manuscript as below.

Methods lines 172-176: We attached descriptive labels to each of the clusters in order to provide interpretability to the clustering results. Means were calculated and examined for each of the 11 variables included in the clustering analysis, and labels were selected to capture clinically relevant themes shared by most of the clinicians in the cluster, and to capture variability between clusters.

2. Why was the initial analysis only done on patient with >10% risk of mortality, with sensitivity analysis on the entire cohort? Why not just do the initial analysis on the entire cohort.

The goal of the behavioral intervention was to increase SIC rates in patients with high risk of mortality. Because clinicians received the intervention only for patients with >10% risk of mortality, including patients with <10% risk of mortality may dilute potential differences between groups. Thus, in this analysis, we restricted the cohort to the population receiving the nudge so that we could isolate differences in clinician responses to the nudge. To analyze whether phenotype response to the intervention was similar in the whole cohort (including patients with lower risk of mortality), the sensitivity analysis was performed on the entire cohort and did not show meaningful differences from the primary analysis. In response to the reviewer’s comment, we have now better clarified this as below: 

Methods lines 131-134: Because clinicians received the intervention only for patients with >10% predicted risk of mortality, our primary analysis only included patients with >10% predicted risk of mortality in order to restrict our cohort to the target population of the intervention.

3. In the abstract (page 2 line 36) please add the word patients’ when explaining the multivariable analysis variables.

We have now included the word patients in our explanations of each of the multivariable analysis variables

Abstract lines 36: Primary analyses were adjusted for patients’ sex, age, race, insurance status, marital status, and Charlson comorbidity index.

4. When explaining the statistics behind the oncologist phenotyping please explain why you decided to focus on patient volume rather than other provider characteristics (main center versus satellite, age, number of years in practice, etc.) Seems like the authors did have an underlying hypothesis when choosing patient volume as the phenotype to test.

In Table 1, a summary of all variables included in the LPA are presented, broken down by clinician cluster. As explained in the response to Reviewer 1, Comment 1, we analyzed differences in all covariates to label clusters. Because patient volume best distinguished all clusters, we labelled clusters using this covariate. 

The reviewer’s point about including provider characteristics such as the number of years in practice is well taken. We have now amended language in our results section to make note of this variation.

Results lines 261-262: This class was comprised of 9 general oncologists, containing 21% of the total clinician population. Of the three classes, these oncologists had the most years in practice (mean [range], 8.42 [3.59, 37.0])

Results lines 270-271: Of the three classes, low-volume specialists had the fewest years in practice (mean [range] 5.26 [2.39, 21.0]), saw the fewest patients per week (mean [SD]: 9.2 [5.6]), had the fewest clinic days per week (mean [SD]: 1.6 [0.9]), saw the highest percentage of new patients per week (mean [SD]: 34% [13.1%]),

Results lines 279-281: Of the three classes, high-volume specialists tended to have an intermediate number of years in practice (mean [range] 7.43 [2.06, 31.5]), saw an intermediate number of patients per week (mean [SD]: 24.3 [5.8]),

As for the reviewer’s point on focusing on practice pattern characteristics rather than provider characteristics, we examined all provider characteristics including number of years in practice, specialty vs general oncology in addition to practice pattern characteristics. After reviewing all covariates, practice pattern covariates (in particular, patient volume) most clearly varied by cluster. Thus, we labelled clusters based on these practice pattern characteristics over provider characteristics. For clarity, all covariates including provider covariates and practice pattern covariates are presented in Table 1 and the results and discussion sections. 

5. Please clarify if the 78 clinicians studied were only in the intervention group of the randomized trial, and if so, shouldn’t there only have been 12,170 patient encounters? If both groups were analyzed shouldn’t there have been more physicians?

The original clinical trial for the behavioral intervention utilized a stepped wedge design. In this trial, 78 total clinicians were originally included in control wedges and then were subsequently randomized in 4-week intervals to the intervention at different times over the course of the trial. 12,170 patient encounters took place during control periods. 14,267 patient encounters took place during intervention periods. Notably, while 78 clinicians in total were included in the original trial, the unit of analysis in the original trial and this analysis was the oncologist-APP dyad because oncologists primarily work 1:1 with APPs and because oncologists and APPs share responsibility for ACPs for patients. Hence, we report phenotypes for the 42 medical oncologist-APP dyads who participated in the trial.

6. As already stated above in the major issues, why were there 3 specialty oncology physicians in the generalists phenotype (how the phenotypes were defined needs to be made more clear).

As explained in the response to Reviewer 1, Comment 1, phenotypes were labelled by calculating the mean of all covariates used in the analysis within each LPA-identified cluster, and subsequently identifying the covariates that most clearly define a cluster. For additional detail please see our response to Reviewer 2 Comment 1 below. Because most clinicians in the group identified in the LPA were generalists, we described this group as generalists to provide a conceptual basis for the identified clusters. 

7. The wording in the response section when describing the difference in response rates between the specialists needs to be a little more careful. The authors were not studying the individual doctor response rates. Rather they were testing whether the patients were listed as having an SIC and comparing that between the doctor phenotypes. It may seem irrelevant, but it is important statistically as it implies that the individual doctors were being tested for how well they responded to the intervention which is not the case. The statistics were done on a patient level.

The reviewer is correct here and we thank the reviewer for pointing out this important distinction. The difference-in-difference analysis measured the ACP rate at the patient-level. The outcome we measured was the change in ACP rate between the pre-intervention period and the intervention period. While the difference in ACP rate does not directly measure the response rate at the physician-level, it nevertheless captures the response to the intervention at the level of the patients who are followed by physicians within the identified cluster. We have clarified this as below:

Results lines 289-296: The probability of a high-risk patient (predicted 180-day mortality >10%) receiving an ACP increased significantly following the intervention among patients receiving care from low-volume specialists and high-volume generalists compared to patients receiving care from high-volume specialists. Among patients receiving care from high-volume specialists, the adjusted probability of a high-risk patient receiving an ACP increased from 2.3% pre-intervention to 7.6% during the intervention period. Among patients receiving care from low-volume specialists, the adjusted probability of ACP increased from 3.1% pre-intervention to 20.7% in the intervention period (adjusted percentage-point difference-in-differences relative to high-volume specialists 12.3, 95% CI 4.3 to 20.3, p=0.003) (Table 2). 

8. Discussion: “our analysis suggests that bandwidth and volume are key drivers of response to interventions intended to improve advance care planning and clinician-patient interaction.” Please be careful here with your wording. Your analysis does not suggest that these items are key drivers, rather it suggests that they “may be a driver of”. Your multivariate model only included one phenotypic characteristic of physicians (all of the other variables were patient variables). Therefore, you have no idea whether patient volume or other provider variables may be the actual driver. Maybe patient volume is associated with another physician variable (like training programs, gender, etc.) that are the actual drivers. This analysis did not look at that and therefore cannot claim this one provider variable is the key driver.

The reviewer is correct in that our analysis does not definitively demonstrate that patient volume and bandwidth are key drivers of response to our intervention. As the reviewer points out, our analysis did not include variables such as history of training programs, or practice location. Of the variables we did include, however, we believe that patient volume are two variables that associate clearly with stronger response to the intervention. 

To our reviewer’s point, we have amended the language in our discussion to better reflect this point of feedback and to also reflect the fact that our analysis was non-exhaustive in terms of provider characteristics. 

Discussion lines 388-391: While this analysis did not exhaustively examine all provider and practice pattern characteristics of these oncology clinicians, our analysis suggests that bandwidth and patient volume may be drivers of response to interventions intended to improve advance care planning and clinician-patient interaction.

9. Discussion: "Targeted deployment of ML-based interventions in the future to clinicians most likely or able to respond, while mitigating alert fatigue or workflow interruptions for clinicians less likely to respond, is a viable strategy for future deployment of ML-based clinician decision support tools." One could also argue that an intervention that cannot be implemented by the physicians that are seeing the majority of the patients is probably not a good intervention. If only the doctors who are seeing the least number of patients can intervene than the majority of patients will not be helped by the intervention.

This feedback exactly highlights the importance of our present study. The original clinical trial investigating this behavioral intervention found a quadrupling in Serious Illness Conversation rates in the intervention periods compared to pre-intervention periods. By studying the underlying heterogeneity in response to this intervention, we have now found that most of the response was driven by a few clinicians. We agree with our reviewer that this intervention may not be effective for generalist oncologists, given the low observed response rate amongst generalists. This may dissuade some general oncology practices from taking up an intervention such as this. However, we would argue that conversely this is an excellent intervention for clinicians in the other clusters that have demonstrated a significant response to the intervention. 

By identifying clusters of physicians and underlying heterogeneity in response to our intervention, we identify groups of clinicians that respond well to the intervention and groups of clinicians that exhibit poorer response to our intervention. Understanding the heterogeneity of response to any particular intervention will allow for nudges to be targeted to the clinicians who are most likely to respond, while avoiding undesired effects like alert fatigue as a result of deploying the intervention to higher-volume clinicians less likely or unable to respond, exactly as our reviewer suggests.

We have included additional discussion on the limitations of this intervention in our revised manuscript as below. 

Discussion lines 399-405: As our present study demonstrates, a small cluster of clinicians may respond strongly to a particular intervention while most clinicians exhibit less response, limiting broad application of the intervention to all clinicians in a practice setting. Targeted deployment of ML-based interventions in the future to clinicians most likely or able to respond, while mitigating alert fatigue or workflow interruptions for clinicians less likely to respond, is a viable strategy for future deployment of ML-based clinician decision support tools.

REVIEWER 2 SPECIFIC COMMENTS

1. The use of LPA is creative to try to define clusters/groups of physicians that respond differently to the intervention. However, this is overall less interpretable and more complex, which is reflected in the discussion. Overall, it seems like the investigators' primary objective is to identify characteristics associated with response. To that end, a logistic regression model across characteristics may be the most helpful tool, and I believe it should be included in the study even if it does not end up as a point of emphasis. Otherwise, conclusions are discussed in the context of clusters whose names are potentially overly simplified (comments regarding this challenge below). Rather than generating logistic regression models summarizing physician features with the LPA, it may be more clear to do so with physician characteristics themselves. It may also reduce some of the challenges with small categories caused by the LPA approach. The overall advantage of using the less-transparent LPA approach feels a bit unclear (and less practically useful).

We thank this reviewer for the thoughtful feedback and consideration of our work. Regarding this reviewer’s major comments, our choice of approach was ultimately informed by the question we sought to answer with this analysis. 

The purpose of this study was to identify heterogeneity in clinician response to the clinician-directed intervention by identifying distinct phenotypes of oncology clinicians and examining their response to the intervention. Our purpose was not to identify specific clinical or sociodemographic characteristics associated with response to a behavioral intervention, as many of these characteristics may overlap with each other. The question we chose to explore specifically interrogates the heterogeneity of response and whether all clinicians who participated in this trial can be grouped into clusters that predict response to the behavioral intervention, as identifying these distinct clusters would allow future developers of interventions to target the interventions towards phenotypes most likely to response while pursuing different strategies for other phenotypes. We sought to use statistical methods to group clinicians into clusters based on underlying similarities across multiple variables or categories rather than identifying single characteristics associated with response. We chose this approach because we did not want to examine the effect of single variables on intervention response rate in isolation, and because an approach utilizing multiple variable regression may identify a set of characteristics associated with significant intervention response, but it would be unclear how best to apply those results if no single physician fulfills each of those criteria. 

We wished to use an approach that captures interactions between variables and identifies groups of clinicians based on underlying similarity across multiple variables. We also believe that the output of such an analysis would also be more clinically relevant, as it is easier to sort clinicians into pre-defined clusters based on their similarity to physicians in the clusters. While there is more than one way to cluster clinicians, we decided to use LPA because it is a hypothesis-free approach to clustering and hence relatively robust to human sources of bias, and because it handles continuous variables well. Latent profile analysis produces clusters based on underlying patterns of similarity without input from study personnel. 

2. I have a couple of questions for clarification - continuous data is on multiple scales (in this study for instance, clinic days/week versus % new patients versus patient encounters/week). Were these data standardized? Were baseline ACP rates ascertainable from Clarity?

We chose to not standardize variables when performing LPA as standardizing variables has no impact on the results of the clustering algorithm. Baseline ACP rates were ascertainable from Clarity and were used to establish pre-intervention ACP rates.

We have clarified this in our revised manuscript as below. 

Methods lines 163-164: 11 variables described in the data section were included in the LPA. These variables were not standardized in the analysis as it has no impact on the results of the clustering algorithm. 

3. The authors used AIC/BIC/entropy approaches to determine the best fit model. More on this decision making process should be discussed (balancing AIC/BIC, etc). AIC/BIC approaches do also have limitations that have been well-discussed in the statistical literature. The concept of "clinical interpretability" should also be discussed further. It is possible that due to the small sample sizes - particularly in the distribution of some characteristics (which result in imbalanced classes) may not allow the generation of highly distinct classes and that the 1 or 2 class models may not be as overfit as the 3 class model.

The AIC, BIC, and entropy of the 2-class model and 3-class model were comparable. We ultimately utilized a 3-class model because the AIC and BLRT favored the 3-class model Additionally, after observing that the 3-class model distinguished between high and low-volume specialty clinicians, we wanted to explore full spectrum of heterogeneity and wanted to make sure that we did not collapse meaningfully different classes into a single class, given that the statistical estimates of prediction error for 2-class and 3-class models are near-equivalent. We have clarified this point as below.

Results lines 225-232: Models with two latent classes and three latent classes were generated. The entropy of the 2-class model and 3-class model were comparable. The 3-class model was selected as the model of best fit by the BLRT (p = 0.010) and because 3-class model had a lower AIC (2678.46 vs. 2689.46) (S1 Table). In addition, this model was reviewed by the first and senior authors for clinical interpretability and chosen because the 3-class model distinguished between high and low volume specialty clinicians. This model was chosen to ensure the model did not collapse potentially meaningfully different classes into a single class given comparable statistical estimates of prediction error between the 2-class and 3-class models.

4. While the results appear to make sense, I think the authors should discuss the limitations of small sample sizes more. Using only 5 oncologists to define a group limits its external generalizability. Only 6 oncologists in the study were generalists, and only 5 were classified into the "low-volume specialists group." While the overall diversity of the general trial (as the authors have highlighted in the discussion) are an overall credit, this also reduces the sizes of each group and makes it more challenging to characterize the subgroups (increasing the brittleness of each group and potential bias). For instance, conclusions drawn on those 6 generalists is highly dependent on those few oncologists; they would also be expected to cluster together as a small group among specialists. Conclusions drawn here may not reflect differences that would be detected if this study was performed exclusively among generalists for instance. I think this limitation may contribute to comment #2 with regards to the models that had fewer classes.

We thank this reviewer for pointing out this important nuance in the discussion of our study. Because our study focuses on heterogeneity in response to an intervention, any results will always be limited by the characteristics of the underlying study population. Our study is no different in that our study population was comprised of all oncologists at a large academic institution, although our clinician sample included a range of specialist and generalist clinicians. While clusters may have been labelled differently in a different sample, there is likely generalizability of our results to other academic oncology institutions.

We believe that our finding of stronger response to behavioral interventions among lower-volume clinicians with higher baseline ACP rates would likely hold true even among community practices given the intuitive reasons of these clinicians having likely having more time and clinical bandwidth to have these conversations, and likely having higher buy-in to ACPs as suggested by higher baseline ACP rates. We also believe that the clusters identified associate well with the sociodemographic and practice pattern characteristics noted in the cluster labels. Consistent with prior literature and clustering best practices, our analysis identified clusters with a minimum cluster size of at least 10% of the entire study cohort which insulate the results of the analysis against inappropriate influence of any single clinician on cluster characteristic. Finally, we believe that the key positive finding of our study is the identification of clusters of clinicians with heterogeneous response to our behavioral intervention. A similar analysis conducted on a different group of clinicians may yield clusters with different features, however we believe other analyses may find those clusters demonstrate meaningful differences in response to the intervention in question. 

We have clarified this limitation as below: 

Discussion lines 425--432: This study has several limitations. First, this trial was conducted within a single tertiary cancer center with limited sample size. The results of our analysis may be influenced by features of individual oncologists who practice at our center and, the results of this study may be difficult to generalize to other settings whose characteristics of oncologists differ from our sample. However, each cluster includes at least 10% of the study population which insulates our results against inappropriate influence of any single clinician on cluster characteristics, and our findings regarding the potential association of patient volume with intervention effectiveness is likely generalizable given the intuitive reasons that lower-volume clinicians likely have more time and clinical bandwidth to have these conversations.

5. On a related note, the "high-volume generalists" category feels like it may be a misnomer - all 6 of the generalists are in this group (which would be expected as a small minority group), but specialists still make up a fair number in the group. Similarly, the small "low-volume specialists" group also has a particularly high baseline ACP rate and fewer years in practice. Limiting the names to specific dimensions loses the resolution/benefit of including all of the variables in the LPA process. Again, logistic regression would help distill some of these features out (such as volume-based metrics).

The reviewer is correct in pointing out that the labels used to describe each cluster do not always perfectly describe each clinician in the cluster. Rather, these labels were designed to capture what we believed to be the most clinically distinctive characteristics of the clinicians together as a group or cluster. Please see our response to Reviewer 1 Comment 1 for further discussion of our methodology in choosing cluster labels.

We believe that our clustering approach with LPA still has several advantages over a logistic regression based approach when attempting to identify oncologist phenotypes. Certain features associated with heterogeneous response to the intervention only emerge when those features are considered together. Please see our response to Reviewer 2 Comment 1 for further discussion of our logic in choosing LPA over logistic regression.

6. I think there may be an error in line 197 - I think that 82% may be miscalculated.

We thank the reviewer for pointing out this error. We have corrected line 206 to 86%. 

Results, line 214-216: We studied 42 oncologist and oncologist-APP dyads in this analysis. Among oncologists, 26 (61.9%) were male and 16 (38.1%) were female. 6 (14.3%) were general oncologists and 36 (86%) were specialty oncologists.

7. Data availability - I agree that the patient data may not be available, though with 42 oncologists analyzed, I feel like deidentified individual level data should be potentially available for sharing, and I encourage the authors to consider exploring this given the PLOS Data policy.

We thank the reviewer for the suggestion. We have now stipulated that we will provide deidentified individual-level data for sharing, per the PLOS Data policy.

8. I think that the "number of oncology clinicians" in Table 1 is a bit misleading as the analysis is based on dyads rather than individual clinicians.

For clarity, we have removed this row in Table 1 and instead focus on highlighting oncology-APP dyads as the main unit of analysis, and clustering on sociodemographic and practice pattern characteristics of the oncologist of each dyad. 

9. While the authors indicate primacy in line 327-329 in the discussion, I don't believe this is necessarily true. There have been a number of studies now investigating clinician trust and use of AI, particularly in the radiology space. Some of those findings have similar findings as in this study - for instance more junior/trainee radiologists are more likely to follow clinical decision support tools/computer aided diagnosis systems. This historical work should be included and placed in context.

As the reviewer points out, there are several papers in the existing literature that surveyed physicians about their attitudes towards machine-learning based clinical decision support tools and artificial intelligence. These studies almost exclusively rely on semi-structured interviews with clinicians, and group the participating physicians into clusters based on review on the interview transcripts. 

We have now contextualized this prior research in our discussion section; however, we believe that our analysis is one of the first to use a hypothesis-free clustering algorithm to identify clusters and demonstrate meaningful variation by cluster in response to a machine-learning based behavioral intervention. 

Discussion lines 367-371: While prior studies have identified groups of clinicians who vary in their surveyed attitudes towards ML-based clinical support tools (28), this is one of the first studies to identify phenotypes of clinician response to an ML-based clinical intervention studied in a randomized controlled trial and demonstrate significant variation in response to the intervention by phenotype.

REVIEWER 3 SPECIFIC COMMENTS:

1. Line 197: Provide a measure of dispersion, perhaps interquartile range or range, for the median number of years in practice.

We have now included the interquartile range for the median number of years in practice. 

Results line 216-217: The median number of years in practice was 7.4 (IQR 5.3,13.0),

2. Line 198: Provide standard deviations for days oncologist spent in clinic and patients seen per week.

We have now included the standard deviations for the number of days in the clinic per week and patients seen per week. 

Results line 217-218: oncologists spent a mean of 2.8 (SD 1.1) days in the clinic per week and saw a mean of 25 (SD 15.2) patients per week.

3. Line 199-200: Provide a measure of dispersion for these medians.

We have included the interquartile ranges for the median percentage of new patients seen per week and median number of encounters per day. 

Results line 221-222: The median percentage of new patients seen per week was 21% (IQR 15.8%, 24.1%), and median number of encounters per day was 9.3 (IQR 8.0,11.5).

Sincerely,

Ravi B. Parikh and Eric H. Li

---

## [Decision Letter · Decision Letter 1]

14 Jan 2022

PONE-D-21-24615R1Oncologist Phenotypes and Associations with Response to a Machine Learning-Based Intervention to Increase Advance Care Planning: Secondary Analysis of a Randomized Clinical TrialPLOS ONE

Dear Dr. Parikh,

Thank you for submitting your manuscript to PLOS ONE. After careful consideration, we feel that it has merit but does not fully meet PLOS ONE’s publication criteria as it currently stands. Therefore, we invite you to submit a revised version of the manuscript that addresses the points raised during the review process. Please pay particular attention to the request to relabel clusters and the request for an additional regression model.

We look forward to receiving your revised manuscript.

Kind regards,

Randall J. Kimple

Academic Editor

PLOS ONE

Reviewers' comments:

Reviewer's Responses to Questions

**Comments to the Author**

1. If the authors have adequately addressed your comments raised in a previous round of review and you feel that this manuscript is now acceptable for publication, you may indicate that here to bypass the “Comments to the Author” section, enter your conflict of interest statement in the “Confidential to Editor” section, and submit your "Accept" recommendation.

Reviewer #1: (No Response)

Reviewer #2: (No Response)

Reviewer #3: All comments have been addressed

2. Is the manuscript technically sound, and do the data support the conclusions?

Reviewer #1: Yes

Reviewer #2: Partly

Reviewer #3: (No Response)

3. Has the statistical analysis been performed appropriately and rigorously? 

Reviewer #1: Yes

Reviewer #2: No

Reviewer #3: (No Response)

4. Have the authors made all data underlying the findings in their manuscript fully available?

Reviewer #1: Yes

Reviewer #2: Yes

Reviewer #3: (No Response)

5. Is the manuscript presented in an intelligible fashion and written in standard English?

Reviewer #1: Yes

Reviewer #2: Yes

Reviewer #3: (No Response)

6. Review Comments to the Author

Reviewer #1: I appreciate the authors responses to my questions and concerns. I have just one minor revision that I think is important for the interpretability of this paper.

1) Although I understand why the authors labelled the physician clusters (as they wanted to make the clusters easier to interpret), I believe by labelling the clusters in the way that they did they actually made their results and analysis more misleading which in my opinion is a graver sin then making the results more difficult to interpret clinically. I would highly recommend that the authors change the labels of the clusters to cluster 1, cluster 2, and cluster 3 and simply explain what the doctors in the clusters tended to have in common rather than labelling them with demographics that the physicians tended to have in common. I think this would be a critical step in making sure that casual readers will better understand the methods and not misinterpret the results.

Reviewer #2: Thank you again for the authors for their hard work, comments, and revisions. I appreciate the effort they've taken in their thoughtful responses, though still have one primary concern around original comments #1 and 5 focused on supplementing their use of LPA to draw the conclusions highlighted in the manuscript.

As the authors have indicated in their response, I think heterogeneity is a reasonable goal, but it feels a little inconsistent with the language used throughout the manuscript which really focuses specifically on the phenotypes themselves (as emphasized in the title and with discussion centered around the specialization/volume characteristics specifically). I think a regression model should hopefully be a limited amount of additional effort for the authors to supplement their current results, and I think is needed to draw the specific conclusions around these characteristics.

I completely agree with the authors that the LPA can capture complex interactions across variables to generate classifications, but for this same reason, the conclusions and naming of the clusters in the manuscript is a bit discordant and oversimplify these same complex interactions. Drawing conclusions around volume and specialization without a more interpretable approach as supplementation (as emphasized in the abstract) risks the reader drawing incorrect conclusions about the data. For example, as stated in comment #5, the high-volume generalist cluster does not clearly show that high-volume generalists have a greater response, but rather there is a phenotypic group (which still has a substantial 33% representation by specialists) has a greater response to the intervention.

I would really suggest to the authors to include simple regression models to help support their conclusions around these characteristics - this is also a common practice in computational health to provide simpler comparator models. Alternatively, I'd suggest the conclusions should otherwise be framed around the heterogeneous classes rather than the simpler specific volume/time availability characteristics.

Reviewer #3: (No Response)

7. PLOS authors have the option to publish the peer review history of their article (what does this mean?). If published, this will include your full peer review and any attached files.

Reviewer #1: No

Reviewer #2: No

Reviewer #3: No

---

## [Author Response · Author response to Decision Letter 1]

23 Feb 2022

Reviewer response attached in files and repasted below. 

February 22, 2022

Randall J. Kimple MD, PhD

Academic Editor

PLOS ONE

Dear Dr. Kimple and Reviewers, 

Thank you for your thoughtful review of our manuscript, “Oncologist Phenotypes and Associations with Response to a Machine Learning-Based Intervention to Increase Advance Care Planning: Secondary Analysis of a Randomized Clinical Trial.” We appreciate your comments and have responded to each of your concerns below. Manuscript revisions are highlighted in bold with page numbers indicating pages in the clean, revised version of the manuscript. 

REVIEWER 1 SPECIFIC COMMENTS 

1. Although I understand why the authors labelled the physician clusters (as they wanted to make the clusters easier to interpret), I believe by labelling the clusters in the way that they did they actually made their results and analysis more misleading which in my opinion is a graver sin then making the results more difficult to interpret clinically. I would highly recommend that the authors change the labels of the clusters to cluster 1, cluster 2, and cluster 3 and simply explain what the doctors in the clusters tended to have in common rather than labelling them with demographics that the physicians tended to have in common. I think this would be a critical step in making sure that casual readers will better understand the methods and not misinterpret the results.

We agree with the review on this important distinction. We have now amended the language throughout our manuscript to reflect this feedback. As suggested by the reviewer, we have now changed the labels of the clusters to cluster 1, cluster 2, and cluster 3, and describe the features shared by clinicians in each cluster, rather than directly labelling a cluster with the descriptive labels as before. Selected changes are shown below.

Abstract lines 41-52: Three oncologist phenotypes were identified: Class 1 (n=9) composed primarily of high-volume generalist oncologists, Class 2 (n=5) comprised primarily of low-volume specialist oncologists; and 3) Class 3 (n=28), composed primarily of high-volume specialist oncologists. Compared with class 1 and class 3, class 2 had lower mean clinic days per week (1.6 vs 2.5 [class 3] vs 4.4 [class 1]) a higher percentage of new patients per week (35% vs 21% vs 18%), higher baseline ACP rates (3.9% vs 1.6% vs 0.8%), and lower baseline rates of chemotherapy within 14 days of death (1.4% vs 6.5% vs 7.1%). Overall, ACP rates were 3.6% in the pre-intervention wedges and 15.2% in intervention wedges (11.6 percentage-¬¬¬¬point difference). Compared to class 3, oncologists in class 1 (adjusted percentage-point difference-in-difference 3.6, 95% CI 1.0 to 6.1, p=0.006) and class 2 (adjusted percentage-point difference-in-difference 12.3, 95% confidence interval [CI] 4.3 to 20.3, p=0.003) had greater response to the intervention.

Results lines 316-322: The probability of a high-risk patient (predicted 180-day mortality >10%) receiving an ACP increased significantly following the intervention among patients receiving care from class 1 and class 2 oncologists compared to class 3 oncologists. Among patients receiving care from class 3 oncologists, the adjusted probability of a high-risk patient receiving an ACP increased from 2.3% pre-intervention to 7.6% during the intervention period. Among patients receiving care from class 2 oncologists, the adjusted probability of ACP increased from 3.1% pre-intervention to 20.7% in the intervention period (adjusted percentage-point difference-in-differences relative to class 3 oncologists 12.3, 95% CI 4.3 to 20.3, p=0.003) (Table 2).

Discussion lines 438-441: In particular, the intervention was associated with a 5.6-fold and 6.7-fold increase in response rates among class 1 oncologists, who consisted primarily of general oncologists with higher patient volumes; and class 2 oncologists, who consisted primarily of specialists with lower patient volumes; compared to class 3 oncologists, who consisted primarily of specialists with higher patient volumes.

REVIEWER 2 SPECIFIC COMMENTS

1. I completely agree with the authors that the LPA can capture complex interactions across variables to generate classifications, but for this same reason, the conclusions and naming of the clusters in the manuscript is a bit discordant and oversimplify these same complex interactions. Drawing conclusions around volume and specialization without a more interpretable approach as supplementation (as emphasized in the abstract) risks the reader drawing incorrect conclusions about the data. For example, as stated in comment #5, the high-volume generalist cluster does not clearly show that high-volume generalists have a greater response, but rather there is a phenotypic group (which still has a substantial 33% representation by specialists) has a greater response to the intervention.

Please see our response to Reviewer 1, point 1. 

I would really suggest to the authors to include simple regression models to help support their conclusions around these characteristics - this is also a common practice in computational health to provide simpler comparator models. Alternatively, I'd suggest the conclusions should otherwise be framed around the heterogeneous classes rather than the simpler specific volume/time availability characteristics.

We thank the reviewer for this thoughtful suggestion. We’ve now included a logistic regression model as a supplemental analysis. We ran this regression model at the level of the patient wedge and measured the association between various clinician practice pattern characteristics and the likelihood of their patients receiving an SIC while adjusting for various patient level characteristics. Consistent with the results of our LPA analysis, we found that in general, patients seeing specialist oncologists were more likely to receive SICs, as were patients seeing clinicians with a higher number of clinic days per week and clinicians seeing a higher number of new patients per week. 

We have added this additional analysis in our manuscript as below. 

Methods lines 206-214: In a secondary analysis, we used logistic regression to measure the impact of various clinician-level variables on the likelihood of a patient receiving an SIC in both the pre-intervention and intervention periods. The logistic regression was conducted at the level of the patient-wedge with the outcome of SIC receipt. Patient covariates included in the model were patient sex, age, race, insurance status, marital status, and Charlson Comorbidity Index. Clinician-level variables included in model were the number of days in clinic per week, percentage of new patients per week, average patients per week, average encounters per day, years in practice, and end-of-life quality metrics (hospice enrollment rate, inpatient death rate, and chemo utilization at the end of life). All analyses were conducted using R version 3.6.0.

Results lines 423-425: In our secondary regression analysis, specialist oncologists, higher number of days per week in clinic, and higher percentage of new patients per week were associated with significantly greater likelihood of conducting an SIC with a patient (S3 Table).

Sincerely,

Ravi B. Parikh and Eric H. Li

---

## [Decision Letter · Decision Letter 2]

1 Apr 2022

Oncologist Phenotypes and Associations with Response to a Machine Learning-Based Intervention to Increase Advance Care Planning: Secondary Analysis of a Randomized Clinical Trial

PONE-D-21-24615R2

Dear Dr. Parikh,

We’re pleased to inform you that your manuscript has been judged scientifically suitable for publication and will be formally accepted for publication once it meets all outstanding technical requirements.

Kind regards,

Randall J. Kimple

Academic Editor

PLOS ONE

Additional Editor Comments (optional):

Reviewers' comments:

Reviewer's Responses to Questions

**Comments to the Author**

1. If the authors have adequately addressed your comments raised in a previous round of review and you feel that this manuscript is now acceptable for publication, you may indicate that here to bypass the “Comments to the Author” section, enter your conflict of interest statement in the “Confidential to Editor” section, and submit your "Accept" recommendation.

Reviewer #1: All comments have been addressed

Reviewer #2: All comments have been addressed

2. Is the manuscript technically sound, and do the data support the conclusions?

Reviewer #1: Yes

Reviewer #2: (No Response)

3. Has the statistical analysis been performed appropriately and rigorously? 

Reviewer #1: Yes

Reviewer #2: (No Response)

4. Have the authors made all data underlying the findings in their manuscript fully available?

Reviewer #1: No

Reviewer #2: (No Response)

5. Is the manuscript presented in an intelligible fashion and written in standard English?

Reviewer #1: Yes

Reviewer #2: (No Response)

6. Review Comments to the Author

Reviewer #1: (No Response)

Reviewer #2: (No Response)

7. PLOS authors have the option to publish the peer review history of their article (what does this mean?). If published, this will include your full peer review and any attached files.

Reviewer #1: No

Reviewer #2: No

---

## [Editor Report · Acceptance letter]

12 May 2022

PONE-D-21-24615R2 

Oncologist Phenotypes and Associations with Response to a Machine Learning-Based Intervention to Increase Advance Care Planning: Secondary Analysis of a Randomized Clinical Trial 

Dear Dr. Parikh:

I'm pleased to inform you that your manuscript has been deemed suitable for publication in PLOS ONE. Congratulations! Your manuscript is now with our production department. 

Kind regards, 

on behalf of

Dr. Randall J. Kimple 

Academic Editor

PLOS ONE